# Singing for People with Advance Chronic Respiratory Diseases: A Qualitative Meta-Synthesis

**DOI:** 10.3390/biomedicines10092086

**Published:** 2022-08-26

**Authors:** Lena Ly, Jennifer Philip, Peter Hudson, Natasha Smallwood

**Affiliations:** 1Faculty of Medicine, Dentistry, and Health Sciences, University of Melbourne, Melbourne, VIC 3010, Australia; 2Centre for Palliative Care, St Vincent’s Hospital, Fitzroy, VIC 3065, Australia; 3Central Clinical School, Monash University, Melbourne, VIC 3004, Australia; 4Vrije University Brussels, 1050 Brussels, Belgium; 5The Alfred Hospital, Melbourne, VIC 3004, Australia

**Keywords:** chronic respiratory disease, chronic obstructive pulmonary disease, interstitial lung diseases, quality of life, qualitative research, self-management, singing

## Abstract

Rationale: Although there remains insufficient evidence regarding singing programs as effective strategies for achieving clinically significant health outcomes, this non-pharmacological intervention appears to be subjectively low-risk and well-tolerated by people with advanced chronic respiratory diseases (CRD). Objective: This study sought to examine and synthesize the current qualitative evidence regarding the experiences of participating in singing for breathing programs by people with advanced CRD. Methods: A meta-synthesis of qualitative data was conducted. Electronic databases (Medline, CINAHL, PsycINFO, and EMBASE) were searched for published qualitative studies reporting the effects of singing programs for adults with advanced CRD and their carers. Primary qualitative data were extracted and analysed, which generated descriptive and analytical themes. Results: Themes identified from seven included studies were: anticipation and reluctance to participate; physical and psychological benefits; new sense of purpose and enjoyment; social connection and achievement; and broad views regarding program structure and content. The themes highlighted changing perspectives before, during and after engaging in the singing program, as participants transitioned from initial anxiety to mastery of their chronic condition as the singing program progressed. Participants, however, raised concerns regarding several singing technicalities, the lack of ongoing support after the singing programs’ conclusion and the social impacts of transitioning the sessions online during the COVID-19 pandemic. Conclusions: This meta-synthesis highlights the positive experiences of people with CRD who participate in singing for breathing programs. Further research, including longitudinal qualitative studies, can provide insight into the acceptability and feasibility of singing programs and inform the broader implementation of the intervention.

## 1. Introduction

Advanced respiratory illnesses are disorders that impact the airways and other structures of the lung [1]. People with lung cancer, chronic obstructive pulmonary disease (COPD) and interstitial lung disease (ILD) [2,3,4] frequently experience progressive, frightening breathlessness, cough and fatigue, which affect their quality of life [5,6,7]. Furthermore, people with advanced chronic respiratory disease (CRD) and their carers experience a high prevalence of loneliness and uncertainty, especially if breathlessness is felt to herald death [3] and thus require both psychological and practical supportive care to cope with their symptoms [7].

The wide range of psychological, social and physical concerns for people with advanced CRD means that a multidisciplinary approach to care is needed to optimally address the needs and wellbeing of patients and their carers [3]. Palliative and supportive care is crucial in addressing unmet and often complex needs, however, many patients do not access these services [8,9]. A lack of communication and interconnectedness has also been identified between respiratory, palliative, and primary care [8]. Widespread recognition of the benefits of a holistic approach involving early access to palliative care, which focuses on symptoms and palliation, is needed in conjunction with disease-directed care [9]. New models of care such as integrated respiratory and palliative care services offer the promise of optimizing symptom management, reducing unscheduled healthcare use and addressing barriers to accessing palliative care [9]. Other important multidisciplinary interventions such as pulmonary rehabilitation are well proven to relieve breathlessness and fatigue and improve exercise tolerance and quality of life [10,11]. Yet, data from an American idiopathic pulmonary fibrosis (IPF) registry revealed low rates of referrals of 727 IPF patients, with only 19% referred [12]. Moreover, only 5–10% of eligible COPD patients complete an entire pulmonary rehabilitation program in Australia [11], and the UK national pulmonary rehabilitation audit identified that 40% of referred patients did not enroll or complete the full program [13]. Therefore, new, accessible approaches to symptom management and supportive care are needed.

Singing has been postulated as an intervention for people with CRD to help reduce symptoms such as breathlessness and enhance the quality of life [7]. In the United Kingdom, singing for lung health programs has evolved from small trials to a national program that focuses on utilizing singing techniques to improve breathing control and posture [14]. Group singing programs, where available, are generally conducted by an experienced singing teacher and delivered in hospital-based or community settings once or twice a week over a minimum period of six weeks [7,15]. While anecdotal experience suggests that participation in these singing programs leads to improvements in wellbeing, evidence regarding effectiveness is limited, arising from a handful of single-center, small, low-quality studies [15,16,17,18,19,20] captured in four recent literature reviews [7,14,21,22]. Importantly, the benefits of singing for lung health programs may not be fully identified in clinical trials, in which the primary outcomes have focused on lung function, mobility, exercise tolerance or walking distance [23]. Additionally, structured clinical questionnaires may fail to reveal the full nature of patients’ experiences or the rich emotional impacts of singing programs [24]. In order to provide greater experiential information regarding any benefits from singing for breathing programs and to understand the perspectives of patients and carers, qualitative methodologies are required.

Qualitative research focuses on enhancing understanding of the lived experience of disease, alongside individuals’ interactions with the healthcare system and their specific choices [25]. Individual qualitative studies to date suggest that singing offers a sense of achievement and self-efficacy for people with CRD [15]. The synthesis of multiple qualitative studies can inform the development, implementation and evaluation of health interventions by integrating findings across different contexts [26]. Meta-synthesis is a useful method for identifying research gaps and translating concepts based on current evidence across the literature [26,27]. This study undertook a meta-synthesis of qualitative data with the aim of collating, synthesizing, and evaluating the current evidence regarding the experiences of singing for people with advanced CRD.

## 2. Methods

### 2.1. Design and Search Strategy

A meta-synthesis of qualitative studies was conducted and reported according to the preferred reporting items for systematic reviews and meta-analyses (PRISMA) principles [28]. Data were synthesized and extracted through tabulation of findings.

Electronic databases (Medline, CINAHL, PsycINFO, and EMBASE) were systematically searched using the terms listed in Appendix A in June 2021 to examine published evidence regarding the effects of singing on people with any advanced CRD and their carers. Titles and abstracts were screened and full-text reviews were conducted against the stated eligibility criteria by LL with discussion with JP and NS using the Covidence systematic review software, Veritas Health Innovation, Melbourne, Australia [29]. JP is a palliative care physician and researcher, NS is a respiratory physician and researcher, and LL is a PhD student. Each member of the team contributed their previous experiences and expertise to the group discussions to strengthen the rigor of the research.

### 2.2. Eligibility Criteria

Studies were included if they had been peer-reviewed and published in English from 1980 onwards. Articles were required to report the experiences of singing for people with advanced CRD using primary empirical qualitative data, which may be in the form of at least one direct quote or observations [30,31]. Included studies reported any qualitative data collection methods and qualitative approaches for data analysis [31]. Quantitative and mixed-methods studies were included if the qualitative data were presented separately in the key findings [30]. Studies reporting only quantitative results and aspects such as the intervention’s generalizability, physical outcomes and measures of the program's effectiveness were excluded.

The quality of included studies was independently assessed using the critical appraisal skills program (CASP) by JP and LL, and disagreements were resolved by discussion [32]. The CASP checklist involved two initial screening questions to determine the qualitative study’s research aim, alongside the appropriateness of the methodology used. Following agreement with these screening questions, the checklist was subsequently consulted to examine in turn, the appropriateness of the research design, recruitment strategy, data collection and rigor of data analysis. The relationship between the researcher and participants was examined to determine potential bias and influence during each study’s formulation of the research question, data collection and sample recruitment. Ethical considerations and clarity, as well as contribution of the research findings to existing knowledge or understanding were also assessed [32]. The value of each study was also determined by identifying if there were discussions regarding the need for further research in new areas and how the findings can be scaled up to different populations and settings [32].

### 2.3. Data Extraction and Analysis

Data extracted from studies included: setting, characteristics of the intervention and participants, qualitative methodology and key findings (Table 1). Direct participants’ quotes regarding the singing interventions were extracted and coded [31]. Data were analysed using a three-stage thematic synthesis in accordance with the methods for the thematic synthesis of qualitative research in systematic reviews guidance [27].

Raw data was initially extracted from the included studies by LL to identify similar meanings and emergent themes. The codes were grouped into descriptive themes, which were ‘data driven’ to describe the perspectives of individual participants’ engagement in the singing sessions [27]. These descriptive themes informed the development of ‘theory driven’ analytical themes to provide a broad understanding of the collective experiences of participants [27]. The themes were categorized to capture the changing nature of participants attitudes before, during and after the singing intervention. Data extraction and development of the themes were discussed and interpreted with JP and NS until consensus regarding accuracy and consistency was reached. Illustrative quotes from participants in different studies are reported to highlight ideas and themes.

## 3. Results

### 3.1. Study Selection

Of 331 articles identified, 72 duplicates were removed, and 237 articles were excluded through title and abstract screening (Figure 1). The full text was reviewed for 22 articles, with 15 excluded as they did not meet eligibility criteria. Seven articles were included in the final sample. No additional studies were identified by hand searching of reference lists from the included studies.

### 3.2. Quality Analysis

The quality appraisal considered all selected studies to be of sufficient quality to be included in the meta-synthesis (Appendix B, Table A1).

### 3.3. Study Population

All included studies explored the experiences of people with COPD (*n* = 7 studies), with only one study including people with ILD [33] (Table 2). No studies referred to people with lung cancer. All studies focused on patients’ views, with none involving carers. All studies arose from one of three countries, including the United Kingdom (UK) (*n* = 4) [15,17,34,35,36], Ireland (*n* = 1) [37] and New Zealand (*n* = 1) [33].

### 3.4. Main Findings

Key findings indicate broad international consistency regarding the essential elements of singing for lung health programs that underpin the evolutionary nature of participants’ experiences and engagement. The meta-synthesis of themes revealed that responses to the program voiced by participants evolved over time, from initial anxiety and hesitation prior to the program beginning, through to perceived changes in physical abilities, social engagement and establishment of new routines during the program. As the intervention progressed, participants’ anxiety gradually declined as they were presented with new opportunities to learn and apply new skills related to singing, to breathing control, and to situations beyond the program. Participants’ anticipation and reluctance transformed into an enhanced sense of self-esteem and boosted confidence, which in turn allowed them to expand their social connections. A sense of eagerness to maintain these benefits most particularly those centered on social connectedness was expressed by participants at the program conclusion. The temporal nature of the changes in relation to the conduct of the singing program forms the map against which these changes became evident (Figure 2 and Table 3).

### 3.5. Theme: Anticipation and Reluctance

#### Before Singing Program

Participants were initially doubtful about joining the intervention due to uncertainties arising from a lack of information regarding the programs:


*‘I think we were all a bit sceptical about it in the beginning. I didn’t understand how it was going to happen.’*
[37]

An inability to sing was the source of concern for some participants’ initial reluctance to engage in the singing programs:


*‘Being honest, I wasn’t going to do it…I couldn’t sing…I had no confidence either.’*
[37]

Some participants reported giving the singing intervention the benefit of the doubt:


*‘I thought anything that might improve my lungs, I was going to try anyway.’*
[37]

### 3.6. Theme: Physical and Psychological Benefits

During the singing program, participants described improvements to their breathing, which helped them to become more ‘proactive’:


*‘… an hour’s continuous exercise here, my lungs feel like they’ve had a better workout than an hour in the gym. This is erm, yeah, I was quite surprised coming here to find out how much of an exercise it was for my lungs.’*
[36]

Participants reported reduced hospital admissions and reliance on the healthcare system as they progressed through the singing program:


*‘Since first time in joining the singing group I have not had to spend time in casualty this winter or spring for COPD.’*
[35]

Furthermore, participants reported that singing had a ‘positive impact’ [17] on their mood:


*‘…it’s an up-lifting thing to do for mental health. We spent quite a lot of time laughing.’*
[34]

Some participants found that singing diverted the focus away from their illness:


*‘It does something to the mind…when you sing you can’t feel sorry for yourself and you don’t think of—of something else. For me when I sing it takes me away to a different level.’*
[33]

### 3.7. Theme: New Sense of Purpose and Enjoyment

As the intervention progressed, participants became more enthusiastic about *‘build [ing] singing into daily routine…to overcome difficulties.’* [17]

Regularity of the program appeared to facilitate certainty of attendance among participants, providing them a sense of joy and an activity to look forward to every week:


*‘I always look forward to Wednesdays [rehearsal day]… Yeah, I always come back on time because that’s my number one [priority]. Oh yes, if I’m going away I make sure I can get back for the choir.’*
[33]

Benefits of the singing programs carried into other aspects of the participants’ lives. Many felt that they had gained a ‘new lease of life’ [33] and they ‘fel [t] more like doing stuff’ [36] every day.

The singing programs also provided participants with the opportunity to expand their knowledge to manage their lung condition and utilize *‘the breathing technique [s] in the gym and everyday life.’* [17]


*‘Being around others with COPD prepares me for the future…helps me to learn more about my illness.’*
[17]

### 3.8. Theme: Social Connection and Achievement

Social connection and achievement were reported by participants both during and after the singing program.

#### 3.8.1. During Singing Program

Participants felt comfortable and able to freely express themselves in the group singing environment, where there was ‘no judgement’ [33] from other members who shared similar experiences of illness and symptoms:


*‘There’s something about coming together as a group and doing an activity as a group and knowing that everybody else has got a lung disease. So, you’re not querying ‘Oh why are they gasping or coughing?’ or whatever.’*
[36]

Participants appreciated developing friendships that arose and were maintained through the program:


*‘The atmosphere has been so welcoming. Everybody helps everybody else, you know? Like a group of friends. Because any time I see them shopping, they go ‘Oh hello see you Tuesday!’ And when you get to seventy, it’s amazing to make even one new friend. However, from this group, I would say if it finished, I would take away maybe two or three really good friends from it anyway.’*
[36]

In addition, the weekly programs reduced feelings of social isolation for some participants by providing them with the opportunity to get ‘out of the house’ [17] and connect to the ‘outside world’ [34]:


*‘Socially certainly, and you know it’s turned into quite a highlight of the week. Actually, you know, meeting up with people and ‘cause we always start gossiping the moment we meet up again, you know we actually start a bit earlier than we’re meant to so we can have a bit of a gossip before we start singing, which is really nice.’*
[36]

Many participants reported the ‘sense of achievement’ [15] they felt from singing together as a group:


*‘I think that the thing that really binds us together really—and the cultural side of it is incidental—is the singing and being together, comradeship. You know companionship. Socialising. That’s us.’*
[33]

Motivational program leaders were recognized as a source of encouragement for participants to overcome their initial anxiety about singing:


*‘The “facilitators” (sic)…were excellent. They made sessions light-hearted as well as instructive. In particular they did well to encourage folk like me, who hadn’t sung since he was 8 years old except at church services, to overcome a natural reluctance to dare to make a noise.’*
[35]

Throughout the program, leaders were crucial for fostering warmth, engagement, a sense of achievement and social connection in the group:


*‘The “facilitators” (sic)…were excellent. They made sessions light-hearted as well as instructive. In particular they did well to encourage folk like me, who hadn’t sung since he was 8 years old except at church services, to overcome a natural reluctance to dare to make a noise.’*
[35]

In response to the transition to online from initially face-to face delivery of the program due to the global COVID-19 pandemic, participants commented on the social impacts following their adaptation to the digitally delivered singing program:


*‘Even online, it’s an up-lifting thing to do for mental health. We spent quite a lot of time laughing. Singing as a group is special.’*
[34]

However, not all comments regarding social connection and achievement were positive. There were several adverse comments made by participants in relation to the social disruption of transitioning the singing program online, as well as abandonment experienced at the programs’ conclusion.

Technical difficulties experienced by participants restricted their uptake and engagement in the online singing program with many commenting that *‘meeting at the [venue] was much more engaging.’* [34]

While *‘physically demanding things are better done in a group’*, group motivation was more personal and palpable face-to-face than online. [34]

#### 3.8.2. After Singing Program

Approaching the end of the program, some participants reported feeling abandoned and:


*‘left out in the cold at the very end with no director or future, felt dumped.’*
[35]

Some participants expressed their desires to maintain the social relationships within the singing group:


*‘I have come to regard the social get together and singing as an important part of my life, which in other circumstances I wouldn’t have got involved in and I intend to help in any way to keep our “choir” going after the end of the project.’*
[35]

### 3.9. Theme: Program Structure and Content

#### During Singing Program

Aside from some participants reporting a preference for the online delivery of the singing programs [34] due to concerns regarding the travelling distance to and from the venue, alongside parking problems and the cold environment in the winter months [35], others seemed to be satisfied with the overall administration and organization of the sessions:


*‘Not having attended any preliminary sessions, I am impressed by service when I phoned for information at almost the last day before a singing session. Joining instructions were concise and complete.’*
[35]

A number of participants were satisfied with the selection of both old and new songs for the singing program [35], which made some people realize that they *‘had forgotten how much [they] enjoyed singing’* [17]. Yet, others expressed their disapproval regarding the lyrics and choices of songs, which were too complicated or were a reminder of past negative experiences [17,35].

## 4. Discussion

To our knowledge, this is the first meta-synthesis of qualitative data to examine the experiences of group singing for people with advanced CRD. In this meta-synthesis studies were identified from three countries (Ireland, UK and New Zealand), key findings indicate the evolving nature of participant experiences and engagement in the singing programs, and associated psychological, social and health benefits. Current delivery of singing programs, however, does not adequately provide participants with the ongoing support required after they conclude. The increasing body of qualitative literature also reveals participants’ feedback on program structure and the impact of virtually delivered singing programs on social connection, which is particularly relevant during the current COVID-19 pandemic. Findings of this meta-synthesis also prompt discussions regarding whether group singing for people with CRD should be identified as a community-based intervention or better integrated with patients’ standard medical treatment.

Participants’ experiences evolved over time, ranging from initial reluctance to later describing many positives experiences (better symptom self-management, achievement, wellbeing and social connection) attributed to participating in group singing. In turn, these experiences were followed by a sense of loss at the end of the program. Whilst reluctance to participate in social activities due to physical limitations has been identified previously in some people with CRD [38], the hesitancy to engage in group singing demonstrated in this meta-synthesis specifically focused on concerns regarding the ability to sing, singing in front of others and limited understanding of how singing may be beneficial.

Past evidence has similarly suggested that people with COPD experience uncertainty and stigma, which negatively impacts their illness perceptions and fosters a reluctance to engage in interventions that can improve quality of life [39]. The experience of uncertainty has been described as disabling for people with CRD, most noticeably during periods from referral to initiation of interventions such as pulmonary rehabilitation [39]. Consistent with the ambivalence regarding the potential value of group singing, some people with COPD have similarly been noted to have a ‘give-it-a-go-but-not-convinced’ attitude towards pulmonary rehabilitation uptake [39]. Such an attitude has been seen to reflect a lack of confidence and skepticism towards pulmonary rehabilitation and its benefits [39]. Recognition that uncertainty is a barrier to participation in health programs is crucial in order to facilitate uptake amongst people with CRD.

The qualitative studies included in this review were not designed to examine possible long-term benefits from group singing interventions for people with CRD. However, the sense of loss and abandonment reported by participants at the conclusion of these programs mirrors recommendations that patients are provided with ongoing support once a health program finishes [40].

A key concept emerging from this study is the place of singing programs and whether they are a component of ‘treatment’ or are better considered a community endeavor. The integration of singing programs into usual medical care is a concept that has received limited attention to date. Although holistic multidisciplinary team care, with communication and coordination between primary and secondary care, is recommended in guidelines for people with CRD [8,9], findings from this review reveal that there is a lack of connection between the delivery of singing interventions and patients’ usual care. By contrast, pulmonary rehabilitation typically involves a diverse health professional team, with good collaboration and connection to both primary and secondary care teams [41]. Furthermore, the pulmonary rehabilitation health team, plays a vital role in guiding and supporting patients to achieve positive health behavior change (such as exercise) in daily life [42]. Singing for breathing programs (where available), however, currently tend to be community-based interventions which lie outside the usual treatment paradigm. Nevertheless, this may be a missed opportunity for promoting broader multidisciplinary collaboration, as well as a key opportunity for a more diverse health team to encourage people with CRD to actively engage with their healthcare, self-management, health literacy and understanding of their condition [43].

Currently, most singing leaders who deliver the singing for breathing programs are not music therapists and may not be working in healthcare. Although singer leaders lack disease-specific knowledge and healthcare experience, they usually possess professional choir-leading skills that are crucial to building group enthusiasm and engagement, which in turn allows participants to benefit physically, psychosocially and musically [44]. It remains unclear whether greater inclusion of healthcare professionals, in addition to increasing referral and interaction with formal health care systems in the delivery of the singing programs, would facilitate or impede the adoption of these self-management techniques amongst patients. There is no evidence regarding the impacts of medicalizing this community-based intervention for people with CRD.

Findings from this meta-synthesis also highlighted how social interactions changed with the rapid switch to online delivery of one singing program during the COVID-19 pandemic [34]. Although the online delivery of singing programs improved mood and enjoyment, some participants reported experiencing difficulties with social interactions in the online format alongside other barriers such as limited digital literacy and technical difficulties [34]. These findings build on the existing literature which indicates that people with COPD may be among the most susceptible to loneliness and least able to access healthcare services online [45]. The delivery of singing programs online assumes participants have both access and capability to utilize technology. Yet, current evidence suggests that many people with COPD are unable to access the internet [45]. Furthermore, the broad changes in societal behaviors over the last twenty years, in which many interpersonal interactions occur online with the growing use of social media and video conferencing, have excluded those who do not have or cannot access the internet [45]. Nevertheless, for socially isolated patients or those living in rural or remote communities (particularly in geographically vast countries), the adaptation of interventions such as pulmonary rehabilitation and singing for lung health programs to an online format following the pandemic has increased accessibility and uptake [46]. Previous reports across the literature indicate greater burden of disease with increasing remoteness for Australians with COPD due to poorer access to and use of health services than people living in metropolitan areas [47]. The need to evaluate the accessibility and coordination of all healthcare services relevant in a rural and remote setting has been recommended [48]. The transition to online delivery of health interventions including programs such as singing programs should avoid further isolating patients who do not have access to digital technology due to poverty, remoteness, lack of infrastructure or discomfort with such innovations [45].

### 4.1. Implications

The findings suggest that the impacts of a community-based singing intervention need to be more sustainable to provide ongoing support at its conclusion, which will ensure easier facilitation and maintenance during the transition to self-management for people with lung diseases [40]. The COVID-19 pandemic has brought many changes to the delivery of healthcare. Future research should focus on the comparison of both in-person and online singing programs to determine whether the same level of social connectedness can be achieved among participants across both modes of delivery. Further work is also required to explore the multifaceted intervention’s responses to supporting patient and carers’ psychological, social and physiological needs. Although there is still low-quality evidence to recommend broader implementation of the intervention, this meta-synthesis provides an indication of participants’ preferences of program delivery and selection of songs, which should be incorporated into the development of singing sessions going forward.

### 4.2. Strengths and Limitations

This meta-synthesis was conducted according to recognized qualitative methodology to enhance the rigor and quality of reporting, which provided an understanding of the perceptions and experiences [31] of singing for people with advanced CRD. The review also minimized the risk of bias by following the PRISMA process [28]. Moreover, direct participant quotes as well as authors’ interpretations [31] were extracted from the studies to ensure that data synthesis was close to the primary data. By broadening the eligibility criteria to include adults over the age of 18 years, this review was also able to capture a diverse group of people with advanced CRD. This meta-synthesis provides valuable data regarding the acceptability and utility of the singing programs, which can inform future implementation, scalability and sustainability of the intervention [49].

A limitation of this meta-synthesis is that only articles in English were included. Cultural barriers may be evident across the findings given that the majority of studies included were based in the United Kingdom. The findings may not be applicable to the delivery and funding of healthcare systems in other countries globally. In order to ensure the singing intervention is able to be broadly translated into care, further work is required across more countries to understand the cultural barriers regarding participant engagement, language and their different preferences for songs. Although rich in qualitative data regarding patients’ experiences with their disease and participation in the singing interventions, the majority of the studies focused upon COPD and only one study recruited people with ILD. Studies included in this review omitted the experiences of singing for people with other advanced chronic respiratory diseases such as lung cancer, as well as their carers. Given that the singing programs involved a self-selected group, participants may have been predisposed to the benefits of the intervention, introducing the potential for bias. Included studies only used self-reporting qualitative methods of data collection such as interviews, focus groups or questionnaires, which do not completely capture participants’ social interactions, dynamics and behaviors. Recognizing these limitations, further work is required to explore the impacts of singing from the perspectives of populations other than people with COPD and using other qualitative approaches such as ethnographic research to observe participation over time.

## 5. Conclusions

Whilst progress has been made towards developing an intervention that addresses the physical and psychosocial wellbeing of people with advanced chronic respiratory diseases, there remains a need to explore the sustainability of benefits beyond the singing program’s conclusion for this population. Singing for lung health programs have only been evaluated in three countries, therefore cultural differences in attitudes and responses to singing and health among people with advanced chronic respiratory diseases and their carers have not been examined. To more fully understand the place of this community-based intervention in health care, future work should longitudinally explore the experiences of patients and carers from multiple countries regarding the benefits of group singing programs for lung health.

## Figures and Tables

**Figure 1 biomedicines-10-02086-f001:**
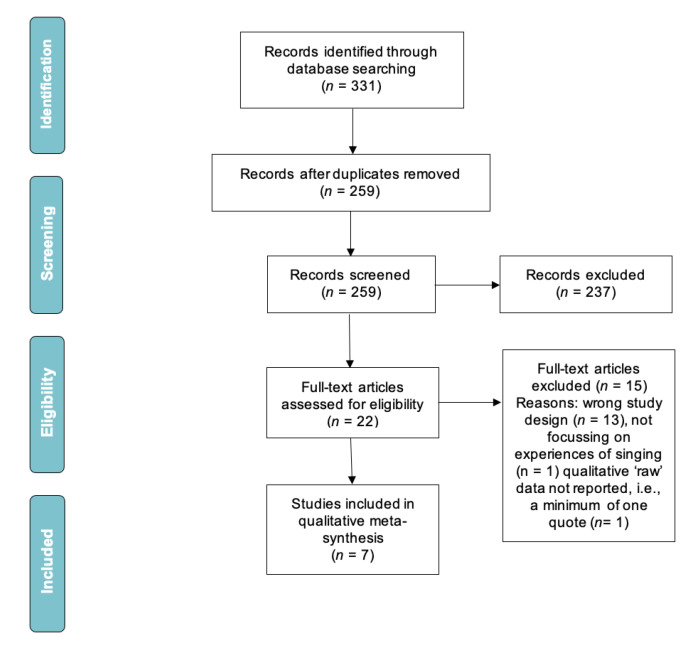
PRISMA flowchart.

**Figure 2 biomedicines-10-02086-f002:**
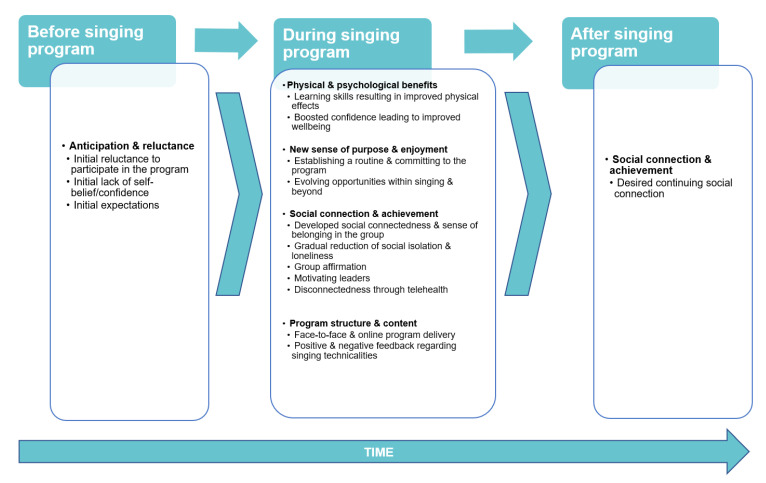
Themes.

**Table 1 biomedicines-10-02086-t001:** Eligibility criteria.

	Inclusion	Exclusion
**Population**	Adults (≥18 years old) with advanced CRD, or specifically people with COPD or ILD and their carers	Singing leaders
**Qualitative data** **collection methods**	Focus group discussions, interviews, observation, etc. [31]	
**Qualitative approaches for data analysis**	Grounded theory, thematic synthesis, framework synthesis, etc. [31]	
**Document type**	Published studies	Duplicate studies, case reports, conference abstracts, literature reviews or documents describing the outcomes of research studies or other primary data

**Table 2 biomedicines-10-02086-t002:** Characteristics of included studies.

Author, Publication (Year)	Aim of Study	Setting	The Singing Program	Year of Participant Recruitment	Participants of the Qualitative Study	Qualitative Approach	Key Findings
**Cahalan et al. (2021)** [37]	To explore participant opinions about a singing intervention: (a) prior to commencement (b) during the program (c) impact on their health and wellbeing.	3 COPD support groups in the Mid-West region of Ireland	8 weekly one-hour singing sessions led by a trained choir leader Groups varied in size (12–35)	May 2019	21 participants (female: *n* = 14 (66%))	Semi- structured focus groups (40 min)	Participants’ feedback was overwhelmingly positive, reporting: improved breathing control efficacy and confidence intervention was fun and enjoyable sessions were a source of social support Preconceptions included: skepticism and anxiety concerns regarding ability to sing Reported negative feedback concerned complicated song lyrics.
**Lord et al. (2010)** [15]	To examine the physical, emotional, and behavioral changes, alongside any detrimental effects or negative experiences of the singing intervention.	Respiratory clinics at the Royal Brompton Hospital, UK	1-h classes held twice weekly for 6 weeks led by a singing teacher 28 people with COPD		Random sample of 8 people with COPD	Interviews (30 min) with a psychologist	Participants’ feedback was overwhelmingly positive, reporting: improved physical performance with increased awareness of breathing control and reduced breathlessness improved general wellbeing increased mood and pleasure developed sense of achievement and self-efficacy No negative experiences were reported.
**Lord et al. (2012)** [17]	To explore the perception of any physical and emotional benefits or harmful experiences of the singing sessions.	Respiratory clinics at Royal Brompton and Harefield NHS Foundation, UK	1-h classes held twice weekly for 8 weeks and led by 1 of 3 singing teachers	April 2010– February 2011	5 participants	Structured interviews (30 min) with a psychologist after the intervention	Participants reported positive feedback: improved breathing techniques and control improved general wellbeing including mood and pleasure developed sense of community and social support No negative experiences were reported.
**McNaughton et al. (2016)** [33]	To explore the health and wellbeing impact of a community singing group on people with COPD.	Community hall in Wellington, New Zealand	Weekly, 1-h sessions led by an amateur singer and respiratory nurse 23 people (13 women and 10 men), 51– 91 years with COPD (*n* = 21) or ILD (*n* = 2)		12 participants	Qualitative description, based on grounded theory and transcripts from individual interviews and a focus group	Perceived health benefits were reported byparticipants including: improvements in breathing, sputum clearance and exercise tolerance improved general wellbeing Key themes emerging from the interviews and focus groups included: being in the right space developed sense of connection with others shared purpose, growth and participation in meaningful physical activity No adverse events were described.
**Philip et al. (2020)** [34]	To determine the overall experience of the singing intervention including the positives, negatives and barriers and facilitators to participation.	Royal Brompton Hospital, London, UK	Weekly, 1-h sessions 18 participants (9 singing and 9 controls) attended 12 sessions		8 of the 9 singing participants	Semi- structured qualitative feedback, telephone interviews. Deductive thematic analysis was used.	Participants reported: singing sessions were enjoyable and beneficial to improving lung condition symptoms and wellbeing preference for face-to-face delivery difficulties achieving psychosocial impacts online due to technological challenges disruption of participation and engagement in personal social interactions online No adverse events were described.
**Skingley et al. (2018)** [36]	To assess the perceived impacts on physical and psychosocial wellbeing among COPD participants.	2 South London boroughs, UK	Weekly, 90 min sessions over 10 months, led by 2 skilled, experienced singing facilitators. Support from a musical director and 2 other facilitators	4–25 April 2016	37 people with COPD	A descriptive qualitative study comprising of interviews (15–30 min) with 4 members of the research team, nested within a single-cohort feasibility study	Participants reported: improved breathing technique improved psychological and social wellbeing No negative impacts were reported.
**Skingley et al. (2014)** [35]	To determine the positive or negative indicators of acceptability, alongside attributions of any changes to wellbeing and changing perceptions over time (at baseline, mid-study and end of study).	Community hall in South East of England	90 min weekly sessions over 36 weeks Singing groups (20–50 people) including patients and supporters: care staff, friends and family). Led by experienced singing leaders	September 2011–June 2012	Comments from 97 individuals with 66 comments at baseline, 77 at mid-study and 73 at final follow-up (total 216 of comments).	A nested qualitative study comprising of questionnaires at baseline, at mid-point (after 5 months) and at the end of the study (after 10 months)	Participants perceived singing as: acceptable beneficial for breathing, general physical, psychological and social wellbeing Positive experiences mainly reported regarding: facilitation and leading organization and administration topic of the research program and content venue and environment ending of the project and future plans Reported negative experiences included: choice of songs warm-up exercises venue, parking and environment abandonment at the program’s conclusion

**Table 3 biomedicines-10-02086-t003:** Meta-synthesis themes.

Time	Themes	Subthemes	Codes
**Before program**	Anticipation and reluctance	Initial reluctance to participate in the program	Skepticism and anxiety, uncertainty
		Initial lack of self-belief/confidence	Inability to sing, lack of a pleasant singing voice, low self-esteem
		Initial expectations	Benefit of the doubt, hopeful for benefits
**During program**	Physical and psychological benefits	Learning skills leading to improved physical effects	Breathing patterns/techniques, reduced breathlessness, posture, functional ability, better exercise tolerance, improved fitness, reduced hospital admissions/healthcare system
		Boosted confidence leading to improved wellbeing	Improved mood and pleasure, fun and enjoyable, motivation to action, laughter and feel-good factor, improved quality of life, distraction from the illness, coping
	New sense of purpose and enjoyment	Established routine and commitment to the program	Integration of singing into daily routine, eagerness, prioritization of program, regularity
		Evolving opportunities within singing and beyond	New opportunities, learning, broader application of breathing techniques
	Social connection and achievement	Developed social connectedness and sense of belonging in the group	Shared growth, experiences, purpose, and responsibility, part of a team, community and support, group mutual understanding, acceptance, sense of ease, lack of judgement, welcoming and comforting atmosphere, freedom to express oneself, developing friendships, commonality
		Gradual reduction of Social isolation and loneliness	Reduction of feelings of embarrassment and isolation, opportunity to leave the house, development and sustainment of friendships, socialization beyond the weekly choir practices, human contact, connection
		Group affirmation	Companionship and comradery, co-construction, motivation, performance, building rapport, disruption to social dynamics, achievement
		Motivating leaders	Support and encouragement, praises, kind leadership, humor, people skills, caring, teaching
		Disconnectedness through telehealth	Less personal nature, difficulty establishing rapport, limited access to digital health and social resources, technical difficulties, limited digital literacy, lack of support, preference for in-person delivery, difficulties with interacting, loudness
	Program structure and content	Face-to-face and online program delivery	Audibility, warmth, physical discomfort, location, amenities, ease of attendance, increase uptake online, personal safety during COVID, duration
		Positive and negative feedback regarding singing technicalities	Repertoire, warm-up exercises, song lyrics, mixed singing ability, song choices, loss of singing technicalities online
**After program**	Social connection and achievement	Desired continuity of the program	Future of the intervention, loss and sudden disconnection

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
