# Peer review of "Singing for People with Advance Chronic Respiratory Diseases: A Qualitative Meta-Synthesis"

_biomedicines, 2022, doi:10.3390/biomedicines10092086_

Round 1
Reviewer 1 Report
The authors have done a careful meta-synthesis of initially 331 published papers in english language about singing for lung health in different medical databases (Medline, CINAHL, PsycINFO, and EMBASE). The focus has been on studies reporting the effects of singing programs for adults with advanced chronic respiratory diseases (CRD) and their carers. Singing has been postulated as an intervention for people with CRD to help reduce symptoms such as breathlessness and enhance quality of life. In the United Kingdom, singing for lung health programs have evolved from small trials to a national program that focuses on utilizing singing techniques to improve breathing control and posture.
Data extracted from studies included: setting, characteristics of the intervention and participants, qualitative methodology and key findings. Data were analysed using a three-stage thematic synthesis in accordance with the methods for the thematic synthesis of qualitative research in systematic reviews guidance. Of 331 articles identified, 72 duplicates were removed, and 237 articles were excluded through title and abstract screening. The full text was reviewed for 22 articles, with 15 excluded as they did not meet eligibility criteria. Seven articles were included in the final sample. The analysis suggests that the physical and psychosocial experiences of participants with advanced lung diseases tend to be positive.
The outhors have carried out an excellent methodological approach for data extraction and development of eligibility criteria. The process of study selection is clearly described and transparency is given for all steps of the study process. Key findings indicate broad international consistency regarding the essential elements of singing for lung health programs that underpin the evolutionary nature of participants’ experiences and engagement. The meta-synthesis of themes revealed that responses to the program voiced by participants evolved over time, from initial anxiety and hesitation prior to the program beginning, through to perceived changes in physical abilities, social engagement and establishment of new routines during the program. As the intervention progressed, participants’ anxiety gradually declined as they were presented with new opportunities to learn and apply new skills related to singing, to breathing control, and to situations beyond the program.
The study highlightens an underestimated type of intervention for patient that suffer from CRD. In this meta-synthesis studies were identified from only three countries (Ireland, UK and New Zealand). This investigation could contribute to take this information about the positive effects of singing for lung health into other countries. It is well done and convincing in its content.
Author Response
Please see the attachment also.
16th August 2022
Prof. Dr. Alice M. Turner
Guest Editor
Biomedicines
Dear Prof. Dr. Turner,
Thank you very much for reviewing our manuscript and providing us with the opportunity to submit our revision. We have addressed all of the constructive questions and comments provided by the reviewers. Details of our changes have been described below each comment and the revisions have been made to the manuscript accordingly via ‘track changes’. Thank you again for facilitating this opportunity.
Reviewer #1:
The authors have done a careful meta-synthesis of initially 331 published papers in english language about singing for lung health in different medical databases (Medline, CINAHL, PsycINFO, and EMBASE). The focus has been on studies reporting the effects of singing programs for adults with advanced chronic respiratory diseases (CRD) and their carers. Singing has been postulated as an intervention for people with CRD to help reduce symptoms such as breathlessness and enhance quality of life. In the United Kingdom, singing for lung health programs have evolved from small trials to a national program that focuses on utilizing singing techniques to improve breathing control and posture.
Data extracted from studies included: setting, characteristics of the intervention and participants, qualitative methodology and key findings. Data were analysed using a three-stage thematic synthesis in accordance with the methods for the thematic synthesis of qualitative research in systematic reviews guidance. Of 331 articles identified, 72 duplicates were removed, and 237 articles were excluded through title and abstract screening. The full text was reviewed for 22 articles, with 15 excluded as they did not meet eligibility criteria. Seven articles were included in the final sample. The analysis suggests that the physical and psychosocial experiences of participants with advanced lung diseases tend to be positive.
The outhors have carried out an excellent methodological approach for data extraction and development of eligibility criteria. The process of study selection is clearly described and transparency is given for all steps of the study process. Key findings indicate broad international consistency regarding the essential elements of singing for lung health programs that underpin the evolutionary nature of participants’ experiences and engagement. The meta-synthesis of themes revealed that responses to the program voiced by participants evolved over time, from initial anxiety and hesitation prior to the program beginning, through to perceived changes in physical abilities, social engagement and establishment of new routines during the program. As the intervention progressed, participants’ anxiety gradually declined as they were presented with new opportunities to learn and apply new skills related to singing, to breathing control, and to situations beyond the program.
The study highlightens an underestimated type of intervention for patient that suffer from CRD. In this meta-synthesis studies were identified from only three countries (Ireland, UK and New Zealand). This investigation could contribute to take this information about the positive effects of singing for lung health into other countries. It is well done and convincing in its content.
We thank the reviewer for their positive review and helpful feedback regarding our manuscript.
Reviewer #2:
I agree that further research should be done to establish the role and place of singing programs applied to patients with chronic lung diseases that significantly affect the quality of life.
We thank the reviewer for their positive review and helpful feedback regarding our manuscript.
Reviewer #3:
The manuscript summarises the results of qualitative studies on singing for people with advanced chronic respiratory diseases. The manuscript is well written, clear, concise. The meta-synthesis approach is particularly appealing. My main concern is related to the fact that numerous reviews/meta-analyses are cited where the original papers would have been preferable (see, for example, ref 7 and 14, cited multiple time instead of the original papers).
We thank the reviewer for their positive review and helpful feedback regarding our manuscript.
Thank you for this suggestion. We note that these references were used for background information to the topic and no data was extracted from these reviews. As such, we politely decline the suggestion to cite the original papers.
I don't know whether table 2 will be reformatted by the publisher, but in its present pdf form it is quite hard to read and in my opinion frankly unacceptable from a graphic point of view.
We apologise for the presentation of the table, which has been corrected. We would greatly appreciate it if the publisher could possibly present all tables in landscape format.
Please double check lines 140-141
We apologise for the error, which has been corrected by removing the phrases ‘was performed’ from the sentence.

Reviewer 2 Report
I agree that further research should be done to establish the role and place of singing programs applied to patients with chronic lung diseases that significantly affect the quality of life.
Author Response

(The authors gave the same response as above.)

Reviewer 3 Report
The manuscript summarises the results of qualitative studies on singing for people with advanced chronic respiratory diseases. The manuscript is well written, clear, concise. The meta-synthesis approach is particularly appealing. My main concern is related to the fact that numerous reviews/meta-analyses are cited where the original papers would have been preferable (see, for example, ref 7 and 14, cited multiple time instead of the original papers).
I don't know whether table 2 will be reformatted by the publisher, but in its present pdf form it is quite hard to read and in my opinion frankly unacceptable from a graphic point of view.
Please double check lines 140-141
Author Response
Please see the attachment also.
16th August 2022
Prof. Dr. Alice M. Turner
Guest Editor
Biomedicines
Dear Prof. Dr. Turner,
Thank you very much for reviewing our manuscript and providing us with the opportunity to submit our revision. We have addressed all of the constructive questions and comments provided by the reviewers. Details of our changes have been described below each comment and the revisions have been made to the manuscript accordingly via ‘track changes’. Thank you again for facilitating this opportunity.
Reviewer #1:
The authors have done a careful meta-synthesis of initially 331 published papers in english language about singing for lung health in different medical databases (Medline, CINAHL, PsycINFO, and EMBASE). The focus has been on studies reporting the effects of singing programs for adults with advanced chronic respiratory diseases (CRD) and their carers. Singing has been postulated as an intervention for people with CRD to help reduce symptoms such as breathlessness and enhance quality of life. In the United Kingdom, singing for lung health programs have evolved from small trials to a national program that focuses on utilizing singing techniques to improve breathing control and posture.
Data extracted from studies included: setting, characteristics of the intervention and participants, qualitative methodology and key findings. Data were analysed using a three-stage thematic synthesis in accordance with the methods for the thematic synthesis of qualitative research in systematic reviews guidance. Of 331 articles identified, 72 duplicates were removed, and 237 articles were excluded through title and abstract screening. The full text was reviewed for 22 articles, with 15 excluded as they did not meet eligibility criteria. Seven articles were included in the final sample. The analysis suggests that the physical and psychosocial experiences of participants with advanced lung diseases tend to be positive.
The outhors have carried out an excellent methodological approach for data extraction and development of eligibility criteria. The process of study selection is clearly described and transparency is given for all steps of the study process. Key findings indicate broad international consistency regarding the essential elements of singing for lung health programs that underpin the evolutionary nature of participants’ experiences and engagement. The meta-synthesis of themes revealed that responses to the program voiced by participants evolved over time, from initial anxiety and hesitation prior to the program beginning, through to perceived changes in physical abilities, social engagement and establishment of new routines during the program. As the intervention progressed, participants’ anxiety gradually declined as they were presented with new opportunities to learn and apply new skills related to singing, to breathing control, and to situations beyond the program.
The study highlightens an underestimated type of intervention for patient that suffer from CRD. In this meta-synthesis studies were identified from only three countries (Ireland, UK and New Zealand). This investigation could contribute to take this information about the positive effects of singing for lung health into other countries. It is well done and convincing in its content.
We thank the reviewer for their positive review and helpful feedback regarding our manuscript.
Reviewer #2:
I agree that further research should be done to establish the role and place of singing programs applied to patients with chronic lung diseases that significantly affect the quality of life.
We thank the reviewer for their positive review and helpful feedback regarding our manuscript.
Reviewer #3:
The manuscript summarises the results of qualitative studies on singing for people with advanced chronic respiratory diseases. The manuscript is well written, clear, concise. The meta-synthesis approach is particularly appealing. My main concern is related to the fact that numerous reviews/meta-analyses are cited where the original papers would have been preferable (see, for example, ref 7 and 14, cited multiple time instead of the original papers).
We thank the reviewer for their positive review and helpful feedback regarding our manuscript.
Thank you for this suggestion. However, we would like to highlight that we have only referenced reviews and meta-analyses in the background section of our manuscript. No data was extracted from these reviews. All data for our meta-synthesis were extracted from original articles.
I don't know whether table 2 will be reformatted by the publisher, but in its present pdf form it is quite hard to read and in my opinion frankly unacceptable from a graphic point of view.
We apologise for the presentation of the table, which has been corrected. The table was submitted in landscape orientation but has been changed to portrait view in the submission process, which reduces the readability of the table. We have removed article titles to help reduce the information in the table. We would greatly appreciate it if the publisher could possibly present all tables in landscape format.
Please double check lines 140-141
We apologise for the error, which has been corrected by removing the phrases ‘was performed’ from the sentence.
Yours sincerely,
Lena Ly on behalf of all authors.
